# Atomistic-Level Description of the Covalent Inhibition of SARS-CoV-2 Papain-like Protease

**DOI:** 10.3390/ijms23105855

**Published:** 2022-05-23

**Authors:** Cécilia Hognon, Marco Marazzi, Cristina García-Iriepa

**Affiliations:** 1Grupo de Reactividad y Estructura Molecular (RESMOL), Departamento de Química Analítica, Química Física e Ingeniería Química, Universidad de Alcalá, Alcalá de Henares, 28801 Madrid, Spain; cecilia.hognon@uah.es; 2Instituto de Investigación Química “Andrés M. del Río” (IQAR), Universidad de Alcalá, Alcalá de Henares, 28801 Madrid, Spain

**Keywords:** SARS-CoV-2, papain-like protease, covalent inhibitor, molecular dynamics, free energy calculations

## Abstract

Inhibition of the papain-like protease (PLpro) of SARS-CoV-2 has been demonstrated to be a successful target to prevent the spreading of the coronavirus in the infected body. In this regard, covalent inhibitors, such as the recently proposed VIR251 ligand, can irreversibly inactivate PLpro by forming a covalent bond with a specific residue of the catalytic site (Cys^111^), through a Michael addition reaction. An inhibition mechanism can therefore be proposed, including four steps: *(i)* ligand entry into the protease pocket; *(ii)* Cys^111^ deprotonation of the thiol group by a Brønsted–Lowry base; *(iii)* Cys^111^-S^−^ addition to the ligand; and *(iv)* proton transfer from the protonated base to the covalently bound ligand. Evaluating the energetics and PLpro conformational changes at each of these steps could aid the design of more efficient and selective covalent inhibitors. For this aim, we have studied by means of MD simulations and QM/MM calculations the whole mechanism. Regarding the first step, we show that the inhibitor entry in the PLpro pocket is thermodynamically favorable only when considering the neutral Cys^111^, that is, prior to the Cys^111^ deprotonation. For the second step, MD simulations revealed that His^272^ would deprotonate Cys^111^ after overcoming an energy barrier of ca. 32 kcal/mol (at the QM/MM level), but implying a decrease of the inhibitor stability inside the protease pocket. This information points to a reversible Cys^111^ deprotonation, whose equilibrium is largely shifted toward the neutral Cys^111^ form. Although thermodynamically disfavored, if Cys^111^ is deprotonated in close proximity to the vinylic carbon of the ligand, then covalent binding takes place in an irreversible way (third step) to form the enolate intermediate. Finally, due to Cys^111^-S^−^ negative charge redistribution over the bound ligand, proton transfer from the initially protonated His^272^ is favored, finally leading to an irreversibly modified Cys^111^ and a restored His^272^. These results elucidate the selectivity of Cys^111^ to enable formation of a covalent bond, even if a weak proton acceptor is available, as His^272^.

## 1. Introduction

Enzyme inhibition is one of the main strategies focusing on disrupting the normal reaction pathway between an enzyme and a substrate. This can be achieved by designing enzyme inhibitors, which are usually molecules of relatively small size compared to the enzyme, characterized by a higher affinity compared to the normal substrate. The development of such inhibitors therefore falls into the category of drug design, constituting one of the main fields in the pharmaceutical industry [1].

From the points of view of physical chemistry and biochemistry, inhibition can be induced following two different strategies: (*i*) non-covalent binding of the inhibitor in the active site of the enzyme; and (*ii*) formation of a covalent bond between the inhibitor and a residue of the enzyme’s active site. As a matter of fact, non-covalent inhibition is a reversible process, which is expected to decrease the enzyme activity without completely blocking it; on the other hand, covalent inhibition is usually an irreversible process, ensuring complete disabling of the enzyme, consequently hampering its function [2].

From the pharmaceutical point of view, non-covalent drugs were preferred until now because of their higher availability and possibility to be repurposed. Nevertheless, their significantly lower specificity can, in principle, increase their possible side effects. On the other hand, covalent drugs were also proposed but to a lower extent because of their inherently more difficult and challenging design [3]. However, covalent inhibitors present much higher specificity and efficacy [4]. 

Medically speaking, there are two main domains of interest for inhibitors’ applications: (*i*) targeting human enzymes, the function of which is involved in some derived illnesses, and (*ii*) targeting viral enzymes, the function of which is usually the evasion of the human immune response, thus favoring the virus proliferation. In this contribution, we will concentrate on this latter domain, considering the interest raised by the infectious COVID-19 disease, which emerged in China in 2019 and then rapidly spread to the whole world [5,6]. 

The virus that causes this infection is the severe acute respiratory syndrome coronavirus-2 (SARS-CoV-2), considering its predecessor (SARS-CoV) in 2003 [7]. The scientific response to the derived global pandemic produced the vaccines now publicly available, although no effective treatments have been discovered; however, a few drugs are promising [8,9]. 

Structurally, SARS-CoV-2 contains a positive-sense single-stranded RNA genome [10]. Its genome is translated to produce all structural proteins like Spike or nucleocapside, and contains two open reading frames (ORFs) responsible for coding the non-structural proteins (NSPs) [11,12], i.e., proteins that should have a certain function that is, however, still under debate. One of them, NSP3, includes the enzyme papain-like protease (PLpro), a key component of the replicase-transcriptase complex, which has an effect in host immune response [13,14]. Hence, PLpro is an attractive antiviral target against several CoVs, including SARS-CoVs and MERS (Middle East respiratory syndrome) -CoVs, and the inhibition of its activity can thus decrease its role in the repression of the host immune response. Indeed, this enzyme suppresses antiviral signaling and inflammation detection, favoring virus replication [15]. 

In these last 2–3 years, several studies attempted to propose compounds to block, at least partially, the SARS-CoV-2 PLpro activity [16,17,18]. Due to the urgency of the global health situation, drug repurposing has been mainly applied, using in silico techniques like molecular docking and classical molecular dynamics (MD) simulation [19,20], as well as in vitro screening using biochemical assays [21,22,23]. Indeed, computational techniques can afford to investigate in a relatively fast cost-effective way a large number of potential compounds by also applying recently developed artificial intelligence algorithms [24,25,26,27,28,29]. 

Although some studies focused on the molecular mechanism of covalent inhibition of the SARS-CoV-2 main protease (3CLpro [30]), including the recently approved drug develop by Pfizer (PF-07321332 or Paxlovid© [31,32]), very few dedicated to PLpro [33,34,35,36,37]. For this reason, we present in this work a detailed computational study to elucidate the molecular basis and factors driving the covalent inhibition of PLpro by a ligand recently tested through experiments, called VIR251 [37].

From the methodological point of view, the formation of a covalent bond (especially if related to additional chemical reactions) requires more sophisticated approaches compared to non-covalent protein binding, since not only is atomistic molecular dynamics mandatory (i.e., coarse graining should be excluded) but also different force fields for each chemical step should be built. Moreover, the calculation of energy barriers of the key reaction steps calls for hybrid quantum mechanics/molecular mechanics (QM/MM) setups, since the electronic structure of the atoms involved in the reaction should be explicitly considered [38]. Alternative methods, as reactive force fields, could also be used to simulate the covalent bond formation of the inhibitor with its target [39]. Nevertheless, although such an approach could be successful when multiple covalent bonds need to be formed in a material matrix (e.g., polymer cross-linking or oxidations of air/surface interface [40]), the required specificity of the ligand for the active (or interaction) site of a certain protein was more properly described by extensive all-atom classical molecular dynamics and eventually QM/MM approaches [41,42,43].

The covalent binding reaction of the VIR251 ligand to PLpro has been already suggested in a previous study [37] by analyzing the obtained crystal structure. An evident S-C bond has been elucidated between the Cys^111^ sulfur atom and one vinyl carbon atom of the inhibitor, leading to the proposal of a Michael addition reaction. However, a detailed description of the steps that are supposed to be followed for a Michael addition has not been performed, it being crucial to the full rationalization and eventual improvement of the inhibition mechanism. 

In particular, Rut et al. [37] already proposed three steps based on experimental findings: *(i)* deprotonation of the nucleophilic thiol group of Cys^111^ by a nearby residue or solvent molecule acting as a Brønsted–Lowry base, forming the anion Cys^111^-S^−^; *(ii)* the consequent nucleophilic addition of Cys^111^-S^−^ to the electrophilic vinyl carbon (C_β_) of VIR251; and *(iii)* the final proton transfer from the previously protonated base toward the covalently bound ligand in its enolate form. Nevertheless, due to the lack of experimentally related data, there is no information about first fundamental step of the mechanism: the binding of the ligand, necessary to be thermodynamically and kinetically favorable, to allow the further formation of a covalent irreversible bond. 

Hence, this work aims to study at the atomistic (and when necessary, electronic) resolution the proposed covalent inhibition mechanism, starting from ligand addition to PLpro (step 1), considering both protonated and deprotonated Cys^111^, expecting an equilibrium between Cys^111^-SH···His^272^ (state A) and Cys^111^-S^−^···^+^H-His^272^ (state B) protein patterns. Such ligand binding and eventual Cys^111^ deprotonation (steps 1 and 2 in Figure 1) constitute a cycle that can lead to covalent inhibition only once [Cys^111^-VIR251]^−^ is formed, followed by proton transfer from the previously protonated His^272^, thus restoring electron neutrality (steps 3 and 4 in Figure 1). 

To elucidate the proposed mechanism, MD equilibrium simulations of each mechanistic step have been performed. Apart from giving an unprecedented insight into this mechanism, the obtained results could aid the design of more selective and efficient covalent inhibitors able to block the PLpro activity and hence the coronavirus infection. The results allowed us to make conclusions about the thermodynamic feasibility of the process, pointing to a significant interaction between the inhibitor and the protease pocket only when Cys^111^ is protonated. We have also analyzed the role of His^272^, in the proximity of Cys^111^, as the base responsible for Cys^111^ deprotonation: both structural MD analysis and energy barrier evaluation at the QM/MM level point toward a much more stable Cys^111^-SH···His^272^ pattern, instead of Cys^111^-S^−^···^+^H-His^272^. Moreover, we have analyzed in detail the effects of the VIR251 covalent binding on the PLpro secondary structure and stability. Finally, we have clarified that ^+^H-His^272^ is the proton source to neutralize the [Cys^111^-VIR251]^−^ moiety, ruling out the participation of the eventual surrounding water molecules. All these results allow the presenting of a general picture of the covalent binding inhibition mechanism in SARS-CoV-2 PLpro, providing structural and energy details about the steps limiting the efficiency of the process. Hence, novel and more efficient covalent inhibitors, built by a de novo design or modification of already reported ones, can be proposed following the conclusions drawn in this work.

## 2. Results and Discussion

In this section, we will study in detail the four steps proposed for the mechanism of PLpro covalent inhibition (Figure 1) by the VIR251 ligand. In particular, the main questions to be addressed are: *(i)* Is the ligand entry to the PLpro a thermodynamically favorable process? *(ii)* Is energetically favorable the Cys^111^ deprotonation by His^272^ and does it influence the stability of the system? *(iii)* Does the covalent binding have a significant influence on the PLpro structure? *(iv)* Which is the proton source for neutralizing the attached ligand, evolved into an enolate—the protonated His^272^ or the surrounding water molecules? In the following, we will propose answers to each of these questions.

### 2.1. VIR251 Ligand Binding to the PLpro Active Site

The first step required for covalent inhibition is the entry of the ligand inside the PLpro pocket. The position of VIR251 inside the protease pocket is known because of its already published X-ray structure [37]. In particular, VIR251 is placed in the S1–S4 pocket of PLpro close to the catalytic Cys^111^ (Figure 2). An MD simulation of this system (i.e., of PLpro with VIR251 non-covalently bound to it) has been performed, confirming the stability of this ligand inside the pocket (Appendix A). Moreover, we have checked the main interactions between VIR251 and the protease pocket, finding similar interactions as the ones previously reported [37]. In particular, hydrogen-bond interactions between VIR251 and Tyr^264^, Gly^271^, Gly^163^, Tyr^268^, and Arg^166^ have been found.

Although the stability of this system is confirmed, it is crucial to evaluate the thermodynamical feasibility of the ligand entrance. For this aim, the ligand-binding free energy and entropy have been calculated by using the MM-PBSA method (see Section 3 and Appendix A), finding that the ligand entry is thermodynamically highly favorable as a free energy of −15.63 kcal/mol has been computed. This spontaneous process points to significant and stable intermolecular interactions between the ligand and the S1–S4 PLpro pocket. Regarding the binding entropy, a negative value of −103.9 cal/mol·K has been found, in line with the negative values expected for the biological systems for which the ligand entry decreases the available microstates, limiting its mobility. From these data, we can estimate the binding enthalpy, considering a temperature of 298.15 K. Its computed value is −46.5 kcal/mol, hence corresponding to an expected exothermic process that counterbalances the entropy-binding loss. 

By analyzing the MD simulation of state A, that is PLpro in absence of VIR251, we observe that the side chain of Cys^111^ is close to the one of His^272^ during the whole simulation time. 

By analyzing the Cys^111^-H···N-His^272^ distance (Figure 3A) along the simulation time, two main patterns are found (Figure 3B). The first one is characterized by a strong hydrogen-bond interaction with an average Cys^111^-H···N-His^272^ distance of 2.1 Å. The second pattern, characterized by an average Cys^111^-H···N-His^272^ distance of ca. 3.7 Å, indicates disruption of the hydrogen bond, mainly due to free rotations of the Cys^111^ thiol group and of the His^272^ imidazole ring. It should be noted that these rotations leading to different Cys^111^ and His^272^ side chain conformations are definitely fast processes, both being patterns found uniformly along the whole simulation time (Appendix A).

Motivated by the fact that this proton transfer could take place independently of the ligand binding, we have also evaluated the thermodynamical feasibility of VIR251 entry once this proton transfer has occurred. That is, we calculated the ligand-binding free energy and entropy considering now Cys^111^ deprotonated and His^272^ protonated (Cys^111^-S^−^···^+^H-His^272^). In this case, the computed binding free energy is slightly positive (1.65 kcal/mol), indicating a non-spontaneous entry of the ligand in the binding pocket. The computed binding entropy is −94.4 cal/mol·K, quite similar to the one calculated for state A (Cys^111^-SH···His^272^). Nevertheless, in this case the binding enthalpy does not counterbalance the entropic loss. Indeed, although the ligand entry is an exothermic process (−26.5 kcal/mol), it is not sufficient to result in a spontaneous process. 

Therefore, VIR251 binding to PLpro should take place prior to Cys^111^ deprotonation, i.e., state C is preferred over state D. To further validate these results, we have performed three sets of MD simulations of state D, that is, PLpro considering Cys^111^ → His^272^ proton transfer, together with VIR251 placed in the S1–S4 pocket. In this particular case, only in two of the three simulations performed is VIR251 kept inside the protease pocket, while in the other one it leaves this pocket after ca. 150 ns (Appendix A), showing an intrinsic and evident low stability. Structural analysis to rationalize this finding will be discussed in the next section. This low stability is in line with the slightly positive binding free energy computed for this system. So, it appears evident that VIR251 binds non-covalently to PLpro prior to Cys^111^ deprotonation, since the ligand tends to leave the protein any time the proton is transferred to His^272^. 

Apart from the thermodynamic evaluation, we have also analyzed the structural influence of VIR251 entry within PLpro. It should be remarked that VIR251 is stably located inside the S1–S4 protease pocket along the simulation time of two independent MD simulations for state C (Appendix A). By comparing the MD trajectories of states A and C, that is, in the absence or presence of VIR251, we can draw some conclusions. At a first glance, a conformational change of the β_14_–β_15_ loop (also known as BL2 loop) upon ligand entry is evident. This was already noticed in a previous work by comparison of their corresponding X-ray structures [37] and is now confirmed by MD. We deeply examine this conformational change along the simulation time by analyzing the loop stability and its closed/open conformation by measuring the distance between Leu^162^ and Tyr^268^ (see Figure 4A for distance definition). The shorter the distance between these two residues, the more closed is the loop conformation, preventing the ligand from leaving the pocket. In this case, it is observed that once the ligand is in the pocket, the flexibility of this loop is decreased, showing a larger stability and so a lower fluctuation of the Leu^162^–Tyr^268^ distance (Figure 4C). This increased stability is partially due to strong hydrogen-bond interactions between VIR251 and amino acids placed both in the mentioned loop (Gly^271^ and Tyr^268^) and in the pocket region opposite to the loop (Gly^163^) as shown in Figure 4B. Moreover, by comparing the Leu^162^–Tyr^268^ distance values along the MD simulations of state A and C, it is evident that the loop is more open in presence of the ligand, with an average distance of ca. 11.6 Å, while it reaches quite close conformations (ca. 6-7 Angstroms) when the pocket is empty, with an average distance of 9 Å. This finding can be reasoned in terms of steric hindrance of the ligand and the loop, forcing a more opened, but at the same time more stable, conformation of the β_14_–β_15_ loop.

### 2.2. Cys^111^ Deprotonation by His^272^

As discussed in the introduction, after ligand non-covalent binding, the next mechanistic step toward covalent inhibition is the Cys^111^ deprotonation by a Brønsted–Lowry base. Hence, it should be elucidated which residue or moiety is acting as the selected base. By analyzing the MD simulation corresponding to state A, it is observed that His^272^ is close to Cys^111^ (3.37 Å on average), as discussed in the previous section (Figure 3). Moreover, no water molecules or additional residue side chains interact with Cys^111^. Hence, we can safely state that His^272^ is the residue responsible for Cys^111^ deprotonation. To confirm this statement, we looked at the Cys^111^···His^272^ interaction, that should be therefore kept once the ligand is inside the S1–S4 PLpro pocket. If, on the contrary, VIR251 disrupts this interaction, other residues or solvent molecules should afford Cys^111^ deprotonation. As a straightforward but effective analysis, we monitored the Cys^111^–His^272^ distance (as defined in Figure 3A) along the MD simulations performed for state C, and compared it with the values obtained for the state A trajectory: it is observed, contrary to what was expected, that the Cys^111^–His^272^ distance decreases once VIR251 is inside the pocket until reaching an average value of 2.3 Å, hence strengthening this hydrogen-bond interaction (Figure 3B). Therefore, we can conclude that although the ligand is placed close to Cys^111^, its deprotonation by the His^272^ side chain should be even more feasible compared to unbound PLpro (state A). By comparing the MD trajectories performed for state A and C, it is clearly observed that for state A, the free rotation around the C–S bond of Cys^111^ leads to conformations with the thiol hydrogen quite far from Hys^272^ (Appendix A). However, such C–S bond free rotation is hampered in state C because of steric hindrance with the ligand, which is placed on top of Cys^111^ (Appendix A). For this reason, the thiol conformations are limited to the ones strengthening the Cys^111^···Hys^272^ hydrogen-bond (Appendix A). 

Once we demonstrated that His^272^ will act as the base, especially in the presence of the ligand, we aimed to evaluate the energetics of the Cys^111^ deprotonation by His^272^. However, no energetic information can be drawn from the classical MD description, it being necessary to introduce an electronic structure theory to describe this chemical reaction. We have especially calculated the proton transfer energy barrier at the QM/MM level (see Section 3), corresponding to ca. 32 kcal/mol (Appendix A), indicating that this proton transfer is in principle feasible, but it is not a fast process. Hence, Cys^111^ deprotonation is a limiting step of the Michael addition overall reaction. Once the Cys^111^-SH···His^272^ → Cys^111^-S^−^···^+^H-His^272^ reaction occurs, the nucleophilic addition to the VIR251 alkene could take place, as will be discussed in the next section. 

Finally, we have analyzed the Cys^111^ deprotonation influence on the system, mainly on VIR251 positioning and the interactions with the protease pocket compared to state C. As aforementioned, the Cys^111^ deprotonation leads to an intrinsic instability of the ligand inside the protease. This finding could be due to weaker interactions between the ligand and the protease pocket, promoting the ligand to move inside the pocket and, at the same time, increasing the flexibility of the BL2 loop. To get some insight, we have compared the hydrogen-bond interactions found between the ligand and the protease pocket for states C and D (i.e., before and after Cys^111^ deprotonation). In both cases, the same residues of the protease pocket are involved in hydrogen-bond interactions with the ligand; however, a great difference arises if we analyze their strengths. In particular, the hydrogen-bonds between the ligand and residues Gly^271^ and Tyr^268^ are almost completely disrupted in state D (Appendix A), which are amino acids forming the mentioned BL2 loop. Although these interactions are severely compromised, novel hydrogen-bond interactions between the ligand and BL2 residues are not observed, leading to a significant instability of the loop conformation compared to state C (Appendix A). The weaker interaction of the ligand with the BL2 loop could be therefore proposed as the main motivation of the observed large conformational changes of the ligand CH_2_-CO-NH group, close to the vinyl moiety, along the simulation time of state D (Appendix A). The rotation around the peptide bond of the ligand is favored for state D because of its lower interactions with the pocket, while for state C the same peptide bond rigid conformation persists during the simulation because of its strong hydrogen-bond interactions with the residues of the BL2 loop and Gly^163^ (opposite to the BL2 loop). This supports the lower stability of the ligand inside the protease pocket after Cys^111^ deprotonation.

Apart from the interactions and stability of VIR251 in state D, it is interesting to analyze the possible role of the ligand in favoring Cys^111^ deprotonation, for instance by leading to a hydrogen-bond interaction with the protonated His^272^, hence stabilizing the proton-transfer process. For this aim, we have analyzed the interactions between VIR251 and H^+^-His^272^ along the simulation time for state D. However, no noticeable interactions were found.

### 2.3. Cys^111^-S^−^ Nucleophilic Addition to VIR251

In this section, we analyze the third step of the covalent inhibition mechanism under study: the nucleophilic addition of the deprotonated Cys^111^-S^−^, generated in the previous step, to the VIR251 C_β_ electrophilic carbon (step 3 in Figure 1). For this reaction to take place, the Cys^111^-S^−^ and the C_β_ atom of VIR251 should be relatively close. To check this hypothesis, we have analyzed the distance between the aforementioned atoms (distance defined in Figure 5A) along the two MD simulations performed for state D, in which VIR251 stays inside the protease pocket. In both cases, the S–C_β_ distance is significantly large compared to the distance equilibrium value of a S–C bond, with an average value of ca. 6.8 Å (Figure 5B). This large distance is not due to ligand moving inside the protease pocket but to the large conformational changes of the CH_2_-CO-NH ligand moiety, as discussed in the previous section. As the ligand does not strongly interact with the BL2 loop, its peptide bonds can partially rotate, sampling conformations with the vinyl methyl ester group (containing the C_β_ atom) quite far from the deprotonated Cys^111^. However, if we analyze the S–C_β_ distance for state C, that is prior to Cys^111^ deprotonation, we observe that they are quite close, being on average ca. 3.9 Å (Figure 5B). Since the nucleophilic addition is possible only once the proton is transferred, we can conclude that not only is the proton transfer a slow process, but, additionally, the nucleophilic addition should take place right after the proton transfer (in an almost concerted fashion); otherwise, the higher flexibility experienced by the ligand will separate the reactive Cys^111^-S^−^ from the VIR251-C_β_ atom, favoring instead the back proton transfer reaction (Cys^111^-SH···His^272^ ← Cys^111^-S^−^···^+^H-His^272^) and thus hampering the covalent inhibition of PLpro.

To check this hypothesis, we have monitored the Cys^111^-S^−^···^+^H-His^272^ distance along the simulations of state D, finding an average value of 2.1 Å (Appendix A). By comparing these values with the Cys^111^-S^−^···C_β_-VIR251 average distance, we can conclude that the -S^−^ moiety is quite closer to H^+^-His (and moreover forming a salt-bridge) than to VIR251-C_β_ (Appendix A). This finding confirms our hypothesis that upon Cys^111^ deprotonation, its reverse process leading to the neutral Cys^111^ would be more probable than nucleophilic addition. Hence, we can conclude that the Cys^111^ deprotonation by His^272^ is a reversible process leading to an equilibrium shifted to the neutral Cys^111^ form, whereas the nucleophilic addition results in a less probable but irreversible process. 

In case the nucleophilic addition takes place, the corresponding enolate is formed (state E in Figure 1). To evaluate the flexibility of this intermediate inside the protease pocket, we have performed two independent MD simulations, observing structural stability (Appendix A). Therefore, we have analyzed the possible sources of stabilization of the negatively charged [Cys^111^-S-VIR251]^−^ enolate offered by the pocket. This stabilization could arise from either the interaction with positively charged amino acids through the formation of salt-bridges or through hydrogen-bond interactions. Since no positively charged residues are close to the enolate group, we can discard electrostatic interactions. Regarding the hydrogen-bond interactions, it is observed that, in state E, two water molecules form hydrogen-bonds with the negatively charged oxygen atom of the enolate moiety (Figure 6A), whereas for state C and D only an interaction with one water molecule is observed (Appendix A). 

### 2.4. Enolate Proton Abstraction

The final step of the PLpro covalent inhibition is the proton transfer from the protein toward the acceptor [Cys^111^-S-VIR251]^−^ to restore the electron neutrality (i.e., formation of Cys^111^-S-VIR251-H, corresponding to state F in Figure 1). In particular, the carbon close to C_β_, where the nucleophilic addition took place, is the atom accepting the proton (C_α_ in Figure 1 and Figure 6A). Hence, a group or moiety able to donate this proton should be located close to this C_α_ atom. To address this question, we have analyzed the surroundings of this carbon atom along the two MD simulations performed for state E. In both cases, we found that the protonated His^272^, which accepted in step 2 the proton of Cys^111^, is now quite close to C_α_, with an average distance of only 1.9 Å (Figure 6B). Hence, a very strong and direct interaction between these two moieties is evident, pointing to ^+^H-His^272^ as the proton-donating group in this step. If we globally analyze the mechanism, the proton accepted by C_α_ is therefore the thiol hydrogen atom initially bound to Cys^111^. Summarizing, the whole mechanism can be explained mainly by the roles of Cys^111^, His^272^, and, of course, VIR251.

Finally, we have checked the dynamic stability of the final state, that is, the neutral VIR251 covalently linked to Cys^111^ (state F in Figure 1). By analyzing the two sets of independent MD simulations, we can conclude that the system is stable along the simulation time (Appendix A) and it keeps the same position in the protease pocket as the one observed for state C, that is, before the nucleophilic addition, locating hydrogen-bond interactions with the same residues (Appendix A). Moreover, as His^272^ has transferred the proton to the ligand, in state F the non-protonated nitrogen atom of its imidazole ring leads to hydrogen-bond interactions with water molecules or Tyr^273^. 

## 3. Methods

### 3.1. Molecular Dynamics Simulations

To characterize the reaction pathway between VIR251 and the active site of PLpro, we performed classical MD on six different systems. Two systems correspond to native PLpro, with either neutral or deprotonated Cys^111^ (states A and B). The other four systems correspond to each mechanistic step (steps C to F, see Figure 1). For this study, we have made use of the PLpro/VIR251 crystal structure available on PDB (PDB code: 6wx4) to build the corresponding systems. For that, three force fields for VIR251 were parameterized (steps 2, 3, and 4) with the amber force field, following the usual amber antechamber procedure. The restricted electrostatic potential (RESP) [44] procedure was used consistently, and the ground state geometry of the three forms of VIR251 were optimized at the density functional theory level using the standard 6-31G basis set and the B3LYP functional [45,46] (see Supporting Information). The different protein/VIR251 models have been solvated, using tleap, in a cubic water box described by using the TIP3P water model [47,48] with the Amber99SB force field [49] applied to the protein. Furthermore, K+ cations were added to ensure electroneutrality of the simulation box. To speed up the simulation, the H mass repartition (HMR) algorithm [50] has been used, in combination with the rattle and shake algorithms [51]. It increases the time step used from 2 ps to 4 ps to integrate Newton’s equations of motion by artificially scaling the mass of all non-water hydrogen atoms from 1.008 to 3.024 Da. Our MD simulations were performed using the NAMD 2.13 code [52]. Equilibrium MD were run, after equilibration and thermalization, in the constant pressure and temperature (NPT) ensemble at 300 K and 1 atm, considering periodic boundary conditions. The protocol used has been the following: 5000 steps of minimization to remove bad contacts followed by 15 ns of dynamics with constraints applied on the protein and the VIR251 molecule. These constraints have been progressively released during these 15 ns. Afterward, production trajectories of 200 ns each have been performed. One MD simulation run has been performed for the systems in absence of ligands (states A and B, in Figure 1), while two MDs have been performed for states C, E, and F, and three MDs were run for state D. The total simulation time for each trajectory is 200 ns. All results have been analyzed and visualized with VMD [53] and Amber Tools [54]. 

### 3.2. Thermodynamic Evaluation of Step 1

To estimate the binding free energy (ΔG) and the binding entropy (ΔS) of the PLpro/VIR251 complex, we used the molecular mechanics/Poisson–Boltzmann surface area (MM/PBSA) approach [55], implemented in AMBER software [54]. In total, 1000 extracted frames, along the two trajectories of systems B and C, were analyzed for calculating the change in Gibbs free energy (ΔG), and 100 frames for calculating a change in entropy (ΔS).

The number of frames analyzed to calculate the thermodynamic properties were chosen according to the computational time required (much larger for entropy than for enthalpy) and to the level of convergence achieved. Indeed, ΔS was calculated considering 50 and 100 frames, not noting significant differences: the result for 50 frames is −31.38 +/− 4.07 kcal/mol (2 days calculation), while the result for 100 frames is −31.18 +/− 3.99 kcal/mol (4 days calculation). Hence 100 frames were considered enough to reach convergence on the ΔS value, within a reasonable computational time.

Based on the equation ΔG = ΔH − TΔS, we also estimated the change in enthalpy (ΔH) of the complex as ΔH = ΔG + TΔS. 

The first term, ΔG, was calculated through the following equation:ΔG_Binding_ = ΔG_Complex_ − ΔG_Receptor_ − ΔG_Ligand_
(1)
where, in our case, the Plpro protein is the receptor and VIR251 the ligand. The second term, ΔS, was calculated using the normal mode analysis [56]. 

### 3.3. QM/MM Reaction Path from State A to B

The energetic barrier corresponding to the Cys^111^ deprotonation by His^272^ (state A to B) has been evaluated at the quantum mechanics/molecular mechanics (QM/MM) level of theory. These calculations have been performed using the LICHEM software package [57,58]. The QM region comprises 16 atoms (containing the CH_2_–SH moiety of Cys^111^ and the imidazole–CH_2_ moiety of His^272^) and was treated at the density functional theory selecting the B3LYP functional [45,46] and 6-31G basis set. A sphere of 15 Å around the QM region was allowed to optimize at each QM step. The deprotonation path was evaluated through the quadratic sting method (QSM) [59], using 27 beads as the initial guess for the replicas along the path. These beads or intermediate geometries connecting the reactant and product were initially guessed by linear interpolation. 

## 4. Conclusions

In this work, we evaluate the four steps of the proposed mechanism for PLpro covalent inhibition by the ligand VIR251. Crucial information has been retrieved from this study by obtaining insights concerning the role of each step on the overall mechanism, as well as the interactions between the ligand and the protease pocket. These fundamental data allowed us to globally rationalize the process and its efficiency. Moreover, the results could aid the design of novel PLpro covalent inhibitors, possibly resulting in an improved mechanism.

In particular, the main results found for this mechanism are the following. First, we point toward the key role of His^272^, which is not only the Brønsted–Lowry base, accepting the proton from the thiol group of Cys^111^, but, it is also responsible for the proton transfer to the [Cys^111^-S-VIR251]^−^ enolate to finally form the neutral covalently linked ligand to Cys^111^ in the last step. So, it is crucial that His^272^ is close to both, Cys^111^ and the C_α_ of the ligand, and that it is not compromised because of other eventual hydrogen-bond interactions with close-by protease residues. However, although the Cys^111^ deprotonation by His^272^ is mandatory for covalent inhibition (otherwise nucleophilic addition cannot happen), we show that when it takes place, the system becomes quite unstable, increasing the flexibility of the ligand that, eventually, can even leave the PLpro pocket. Hence, due to the close interaction between Cys^111^ and His^272^, also after proton transfer, back proton transfer is in principle possible, reaching a Cys^111^-SH*···*His^272^⇔ Cys^111^-S^−^*···*^+^H-His^272^ equilibrium largely displaced toward the neutral Cys^111^ side chain. Hence, this step was identified as the one most probably limiting the overall inhibition mechanism.

Moreover, we see that due to the already mentioned system instability after Cys^111^ deprotonation, the distance between the -S^−^ nucleophile and the -C_β_ electrophile is quite large after system relaxation, hampering the nucleophilic addition essential for covalent inhibition. Hence, as this is a mandatory step to reach irreversible inhibition, we conclude that this covalent bond formation should occur right after Cys^111^ deprotonation (or, alternatively, in a concerted fashion), as otherwise the two reacting atoms would be too distant. This nucleophilic step is certainly an irreversible step, generating the enolate form of the ligand.

Finally, the neutral form of the covalently bonded ligand to Cys^111^ is possible by proton transfer from ^+^H-His^272^ to the enolate, resulting in a quite favorable process because of the significantly short His^272^–C_α_ distance. So, we can conclude that this step is also irreversible.

To sum up, we can conclude that the overall mechanism is formed by four steps. The first one, the ligand non-covalent bonding to PLpro, is thermodynamically favorable because of its negative free-energy difference. The second step is actually an equilibrium that is largely shifted toward the undesired non-deprotonated Cys^111^. However, the third and fourth steps are irreversible, driving the covalent inhibition.

Hence, the design of novel and more efficient covalent inhibitors for PLpro should promote the Cys^111^ deprotonation, ensuring the stability of the system to keep the electrophilic atom of the ligand close to the nucleophile for covalent bonding. For this aim, the structure of VIR251 could be modified to ensure strong interactions between the ligand and the protease pocket residues after Cys^111^ deprotonation. A second approach would be the introduction of other bases able to accept the thiol proton of Cys^111^ instead of His^272^. Motivated by the close position of the ligand and Cys^111^, the ligand structure could therefore be modified to include a group acting as a strong base to overcome the intrinsic instability of the system after Cys^111^ deprotonation by His^272^. We hope that these results will thus lead to design improvements for PLpro covalent inhibition.

## Figures and Tables

**Figure 1 ijms-23-05855-f001:**
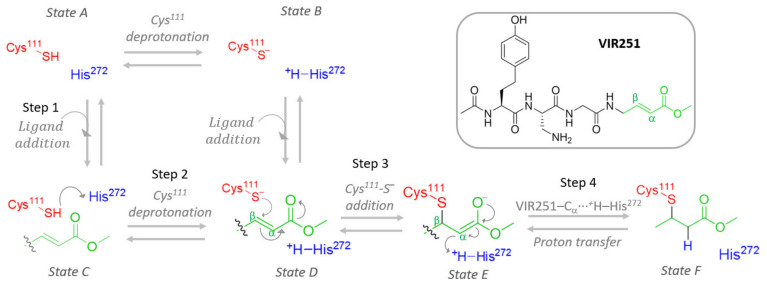
Schematic representation of the four-step mechanism proposed to inhibit PLpro through the ligand VIR251. Two possible protonation states of PLpro (states A and B) and four consecutive steps to reach covalent inhibition (states C–F) are depicted.

**Figure 2 ijms-23-05855-f002:**
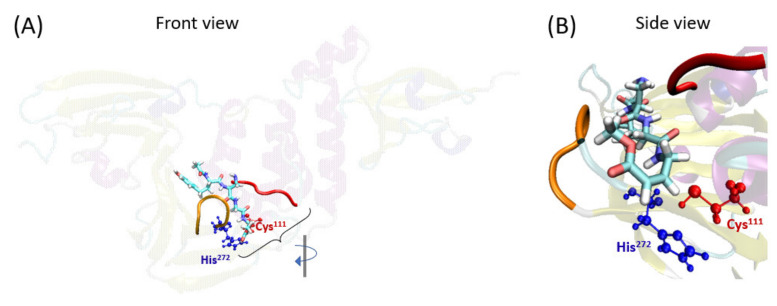
Representative snapshot (111.2 ns) of the ligand VIR251 inside the PLpro S1–S4 pocket (state C). (**A**) Front view: whole enzyme-ligand system, highlighting the active site. (**B**) Side view: zoom into the ligand pocket. In particular, PLpro is depicted using the secondary structure representation, VIR251 is represented in licorice while Cys^111^ and His^272^ as the CPK drawing method. The orange peptide chain represents the BL2 loop (including residues 267 to 271), while the red chain represents the loop in front of BL2 loop, including residues 161 to 164.

**Figure 3 ijms-23-05855-f003:**
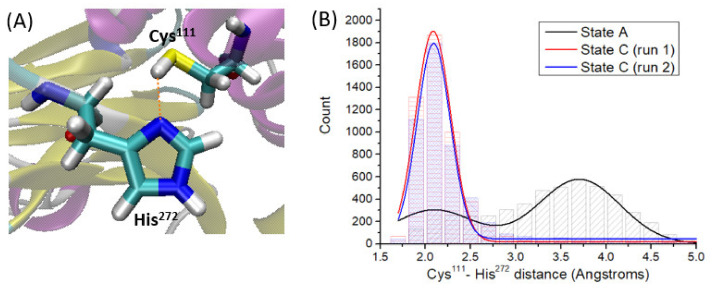
(**A**) Cys^111^-His^272^ definition of the proton transfer coordinate (snapshot extracted at 132 ns). (**B**) Histogram showing the number of snapshots along each trajectory (count) as a function of the Cys^111^-His^272^ distance, measured for states A and C. The respective fitted Gaussian functions are depicted as lines. The two patterns for state A (black line) correspond to the maxima at ca. 2.1 and 3.7 Å.

**Figure 4 ijms-23-05855-f004:**
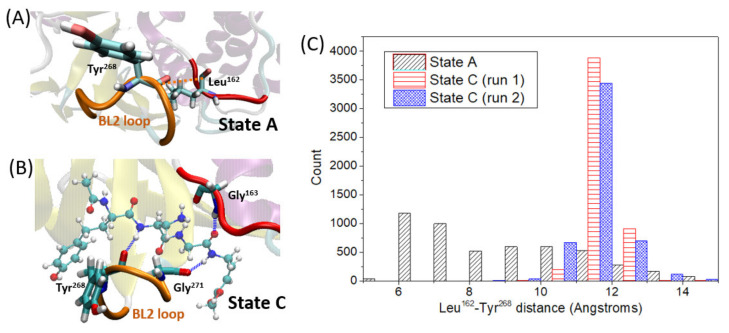
(**A**) Definition of the Leu^162^–Tyr^268^ distance, shown for state A (snapshot corresponding to 65 ns). (**B**) Representative snapshot (111.2 ns) of state C showing hydrogen-bond interactions between the ligand (CPK representation) and residues Gly^271^ and Gly^163^. (**C**) Histogram showing the number of snapshots along each trajectory (count) as a function of the Leu^162^–Tyr^268^ distance, measured for states A and C.

**Figure 5 ijms-23-05855-f005:**
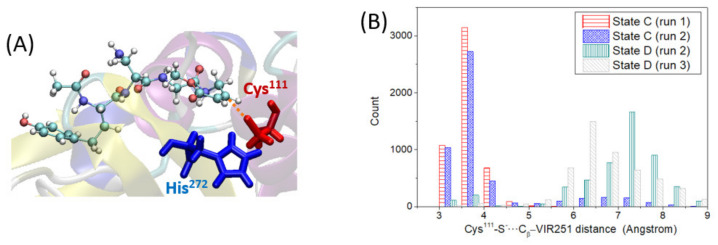
(**A**) Representative snapshot (59.5 ns) of state D, defining the S–C_β_ distance to be analyzed. (**B**) Histogram of the S–C_β_ distance for each simulation performed for states C and D.

**Figure 6 ijms-23-05855-f006:**
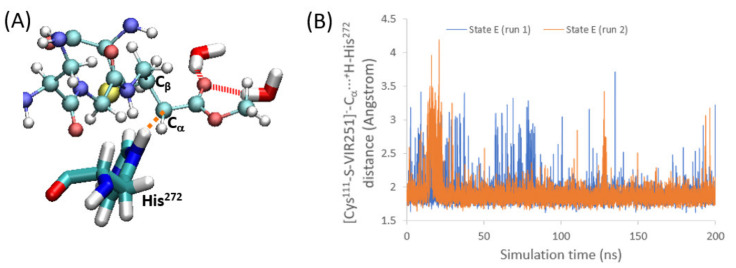
(**A**) Representative snapshot (45.2 ns) of state E, showing the [Cys^111^-S-VIR251]^−^-C_α_···^+^H-His^272^ distance (orange dotted line) and the hydrogen-bond interactions (red dotted lines) between the oxygen atom of the enolate with two water molecules. (**B**) VIR251-C_α_···^+^H-His^272^ distance measured along the simulation time for the two independent trajectories performed for state E.

## Data Availability

Not applicable.

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
