# Peer review of "Atomistic-Level Description of the Covalent Inhibition of SARS-CoV-2 Papain-like Protease"

_ijms, 2022, doi:10.3390/ijms23105855_

Round 1

Reviewer 1 Report

The author reported MD simulations and QM/MM calculations of PLpro pocket is thermodynamically favorable only when considering the neutral Cys111 that is, prior to the Cys111 deprotonation. Due to Cys111-S- negative charge redistribution over the bound ligand, proton transfer from the initially protonated His272 is favored, finally leading to an irreversibly modified Cys111 and a restored His272. These results elucidate the selectivity of Cys111 to enable the formation of a covalent bond, even if a weak proton acceptor is available, as His272. This Manuscript is well written and needs minor revision.

Comments:

  1. Every section of the manuscript must be written scientifically according to the published literature with appropriate references.
  2. The logical flow of this manuscript is not perfect. The authors have written several matters haphazardly. The work appears as groundwork.
  3. Spacing, punctuation marks, grammar, and spelling errors should be reviewed wholly.
  4. The flow of the introduction is not complete and unspecific. My recommendation is to construct the sentences more lucid and legible for more productive comprehension.
  5. The first paragraph of the introduction section contains no new information. Need to change.
  6. The author has mentioned four steps mechanism, ligand non-covalent bonding to PLpro, is thermodynamically favorable due to its negative free energy difference. The second step is actually, an equilibrium largely shifted toward the undesired non-deprotonated Cys 111. However, the third and fourth steps are irreversible, driving the covalent inhibition. What is the significance of the steps four steps mechanism?

Reviewer 2 Report

The Authors report the description of cavalent inhibition of VIR251 towards papain-like protease. The manuscript is quite compelling from idea to experiments. Themanuscript deserves to be published afeter the following revisions:

1) In Figures 2-6 the authors refers to representative snapshot, but the caption lacks the simulation time relative to the represented frame.

2) More details should be reported about equilibration procedure for md.

3) In Figure 3 and 4, what do the authors mean for count?

4) In Figure 2 two views are redundant. Moreover, the Figure is not clear.

5) At page 4, line 163, the authors asses "an evident hydrogen bond", even though the distance is 3.37. It should be revised the assertion.

6) TIP3P is a force field?

7) For Thermodynamic evaluation of step 1, the authors used 1000 extracted frame for enthalpy and 100 frame for entropy. Why?

Reviewer 3 Report

The paper " Atomistic-level description of the covalent inhibition of SARS- 2 CoV-2 papain-like protease" by Hognon et al. proposes by means of MD simulations and QM/MM calculations the inhibition mechanism of the irreversible ligand VIR251 against PLpro. The study is based on the known crystal structure of the PLpro/VIR251 complex. The reaction pathway is dissected in steps that have been analysed separately, evaluating the thermodynamic stability and performing MD simulations. The study comes to conclude that the reaction mechanism occurs in 4 steps proposing that Cys111 has to be protonated to favor the ligand entry in the catalyitic pocket and that the rate-limiting step is the second one, which is a reversible equilibrium shifted toward the unwanted intermediate. The conclusions allow giving hints for the design of more efficient covalent inhibitors.

The paper touch on a very relevant topic, albeit not trivial. Indeed targeting PLpro is resulting to be more difficult than expected as only a few ligands have been identified, despite the huge effort in place. The manuscript is is very well written and concepts and results are clearly exposed. The methods applied are robust and conclusions are convincing. I'm therefore very favorable to proposing the publication of this manuscript with minor improvements and corrections as listed below:

1) Lines 55-56 The following sentence: "Due to the urgency of global health situation, drugs repurposing has been mainly applied, using molecular docking and classical Molecular Dynamics (MD) simulation techniques [19,20]", does not mention the many biochemical screenings that have been performed. This gives the idea that none has been done. Please rephrase mentioning this and the referring literature (you can find some hints in the recent Review: Calleja et al. 2022 10.3389/fchem.2022.876212).

2) Lines 68-69: The sentence "Especially, very few studies focused on the molecular mechanism of covalent inhibition of the SARS-CoV-2 main protease (3CLpro) or PLpro [25–29]" is not entirely true. Indeed this is true for PLpro but not for MPro, where there are several irreversible inhibitors, including the Pfizer compound PF recently approved in clinics (ref. ).

Line 263: the word "instability" is misspelled.

Figure S10 and S11 are mentioned in the main text but are missing in the supplementary file. Please integrate this lack.

In Bibliography there is an extra work "other" before the titles of many references. 

Round 2

Reviewer 1 Report

The revised version of the manuscript includes all remarks and modifications indicated. The main concerns of the manuscript have been solved. In my opinion, the provided version is now suitable for publication

Reviewer 2 Report

The authors responded to all issues raised by this Reviewer.

Reviewer 3 Report

The authors have fulfilled all the required integrations and I'm fully satisfied with the presented version. I'm supporting the publication as it is.